# Chaos Theory and Adversarial Robustness

## Abstract

Neural networks, being susceptible to adversarial attacks, should face a strict level of scrutiny before being deployed in critical or adversarial applications. This paper uses ideas from Chaos Theory to explain, analyze, and quantify the degree to which neural networks are susceptible to or robust against adversarial attacks. To this end, we present a new metric, the "susceptibility ratio," given by $\hat{\Psi}(h, \theta)$, which captures how greatly a model's output will be changed by perturbations to a given input.

Our results show that susceptibility to attack grows significantly with the depth of the model, which has safety implications for the design of neural networks for production environments. We provide experimental evidence of the relationship between $\hat{\Psi}$ and the post-attack accuracy of classification models, as well as a discussion of its application to tasks lacking hard decision boundaries. We also demonstrate how to quickly and easily approximate the certified robustness radii for extremely large models, which until now has been computationally infeasible to calculate directly.

## 1 Introduction

The current state of Machine Learning research presents neural networks as black boxes due to the high dimensionality of their parameter space, which means that understanding what is happening inside of a model regarding domain expertise is highly nontrivial, when it is even possible. However, the actual mechanics by which neural networks operate - the composition of multiple nonlinear transforms, with parameters optimized by a gradient method - were human-designed, and as such are well understood. In this paper, we will apply this understanding, via analogy to Chaos Theory, to the problem of explaining and measuring susceptibility of neural networks to adversarial methods.

It is well-known that neural networks can be adversarially attacked, producing obviously incorrect outputs as a result of making extremely small perturbations to the input (Goodfellow et al., 2014; Szegedy et al., 2013). Prior work, like Shao et al. (2021); Wang et al. (2018) and Carmon et al. (2019) discuss "adversarial robustness" in terms of metrics like accuracy after being attacked or the success rates of attacks, which can limit the discussion entirely to models with hard decision boundaries like classifiers, ignoring tasks like segmentation or generative modeling (He et al., 2018). Other work, like Li et al. (2020) and Weber et al. (2020), develop "certification radii," which can be used to guarantee that a given input cannot be misclassified by a model without an adversarial perturbation with a size exceeding that radius. However, calculating these radii is computationally onerous when it is even possible, and is again limited only to models with hard decision boundaries.

Gowal et al. (2021) provides a brief study of the effects of changes in model scale, but admits that there has been a dearth of experiments that vary the depth and width of models in the context of adversarial robustness, which this paper provides. Huang et al. (2022a) also studies the effects of architectural design decisions on robustness, and provides theoretical justification on the basis of deeper and wider models having a greater upper bound on the Lipschitz constant of the function represented by those models. Our own work's connection to the Lipschitz constant is discussed in Appendix C. Wu et al. (2021a) studies the effects of model width on robustness, and specifically discusses how robust accuracy is closely related to the perturbation stability of the underlying model, with an additional connection to the local Lipschitzness of the represented function. Our experimental results contradict those found in these papers in a few places, namely as to

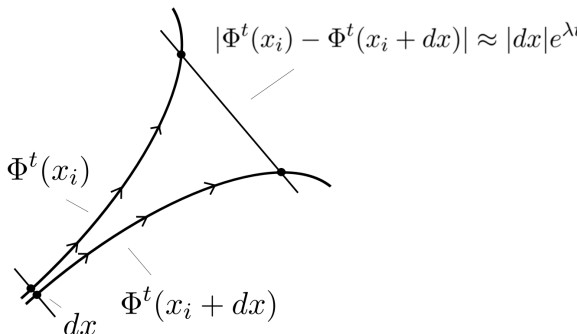

Figure 1: In a dynamical system, two trajectories with similar starting points may, over time, drift farther and farther away from one another, typically modeled as exponential growth in the distance between them. This growth characterizes a system as exhibiting "sensitive dependence," known colloquially as the "butterfly effect," where small changes in initial conditions eventually grow into very large changes in the eventual results.

the relationship between depth and robustness. Additionally, previous work is limited to studying advanced State-of-the-Art CNN architectures, which introduces a number of effects that are never accounted for during their ablations.

Regarding the existence of adversarial attacks *ab origine*, Pedraza et al. (2020) and Prabhu et al. (2018) have explained this behaviour of neural networks on the basis that they are dynamical systems, and then use results from that analysis to try and classify adversarial inputs based on their Lyapunov exponents. However, this classification methodology rests on loose theoretical ground, as the Lyapunov exponents of a single input must be relative to those of similar inputs, and it is entirely possible to construct a scenario wherein an input does not become more potent a basis for further attack solely because it is itself adversarial.

In this work, we re-do these Chaos Theoretic analyses in order to understand, not particular inputs, but the neural networks themselves. We show that neural networks are dynamical systems, and then continuing that analogy past where Pedraza et al. (2020) and Prabhu et al. (2018) leave off, investigate what neural-networks-as-dynamical-systems means for their susceptibility to attack, through a combination of analysis and experimentation. We develop this into a theory of adversarial susceptibility, the "susceptibility ratio" as a measure of how effective attacks will be against a neural network, and show how to numerically approximate this value. Returning to the work in Li et al. (2020) and Weber et al. (2020), we use the susceptibility ratio to quickly and accurately estimate the certification radii of very large neural networks, aligning this paper with prior work.

## 2   Neural Networks as Dynamical Systems

We will now re-write the conventional feed-forward neural network formulation in the language of dynamical systems, in order to facilitate the transfer of the analysis of dynamical systems back to neural networks. To begin with, we first introduce the definition of a dynamical system, per standard literature (Alligood et al., 1998).

### 2.1   Dynamical Systems

In Chaos Theory, a dynamical system is defined as a tuple of three basic components, written in standard notation as $(T, X, \Phi)$. The first, $T$, referred to as "time," takes the form of a domain obeying time-like algebraic properties, namely associative addition. The second, $X$, is the state space. Depending on the system, elements of $X$ might describe the positions of a pendulum, the states of memory in a computer program, or the arrangements of particles in an enclosed volume, with $X$ being the space of all possibilities thereof. The final component, $\Phi : T \times X \to X$, is the "evolution function" of the system. When $\Phi$ is given

a state $x_{i,t} \in X$ and a change in time $\Delta t$, it returns $x_{i,t+\Delta t}$, which is the new state of the system after $\Delta t$ time has elapsed. The $x_{i,t}$ notation will be explained in greater detail later. We will write this as

$$x_{i,t+\Delta t} = \Phi(\Delta t, x_{i,t})$$

In order to stay well defined, this has to possess certain properties, namely a self-consistency of the evolution function over the domain $T$. A state that is progressed forward $\Delta t_a$ in $T$ by $\Phi$ and then progressed again $\Delta t_b$ should yield the same state as one that is progressed $\Delta t_a + \Delta t_b$ in a single operation:

$$\Phi\big(\Delta t_b, \Phi(\Delta t_a, x_{i,t})\big) = \Phi(\Delta t_a + \Delta t_b, x_{i,t})$$

Relying partially on this self-consistency, we can take a "trajectory" of the initial state $x_{i,0}$ over time, a set containing elements represented by $\big\{\big(t, \Phi(t, x_{i,0})\big)\big|\forall t \in T\big\}$. To clarify; because each element within $X$ can be progressed through time by the injective and self-consistent function $\Phi$, and therefore belongs to a given trajectory,[1] it becomes both explanatory and efficient to denote every element in the same trajectory with the same subscript index $i$, and to differentiate between the elements in the same trajectory at different times with $t$. In order to simplify the notation, and following on from the notion that the evolution of state within a dynamic system over time is equivalent to the composition of multiple instances of the evolution function, we will write the elements of this trajectory as

$$\Phi(t, x_{i,0}) = \Phi^t(x_i) = x_{i,t}$$

with an additional simplification of notation using $x_i = x_{i,0}$, omitting the subscript $t$ when $t = 0$.

From these trajectories we may derive our notion of chaos, which concerns the relationship between trajectories with similar initial conditions. Consider $x_i$, and $x_i + \delta x$, where $\delta x$ is of limited magnitude, and may be contextualized as a subtle reorientation of the arms of a double pendulum prior to setting it into motion. We also require some notion of the distance between two elements of the state space, but we will assume that the space is a vector space equipped with a length or distance metric written with $|\cdot|$, and proceed from there. For the initial condition, we may immediately take

$$|\Phi^0(x_i) - \Phi^0(x_i + \delta x)| = |\delta x|$$

However, meaningful analysis only arises when we model the progression of this difference over time. In some systems, minor differences in the initial condition result in negligible effect, such as with the state of a damped oscillator; regardless of its initial position or velocity, it approaches the resting state as time progresses, and no further activity of significance occurs. However, in some systems, minor differences in the initial condition end up compounding on themselves, like the flaps of a butterfly's wings eventually resulting in a hurricane. Both of these can be approximately or heuristically modeled by an exponential function,

$$|\Phi^t(x_i) - \Phi^t(x_i + \delta x)| \approx |\delta x|e^{\lambda t}$$

In each of these cases, the growing or shrinking differences between the trajectories are described by $\lambda$, also called the Lyapunov exponent. If $\lambda < 0$, these differences disappear over time, and the trajectories of two similar initial conditions will eventually align with one another. However, if $\lambda > 0$, these differences increase over time, and the trajectories of two similar initial conditions will grow farther and farther apart, with their relationship becoming indistinguishable from that of two trajectories with wholly different initial conditions. This is called "sensitive dependence," and is the mark of a chaotic system.[2] It must be noted, however, that the exponential nature of this growth is a shorthand model, with obvious limits, and is not fully descriptive of the underlying behavior.

---

[1] Multiple trajectories may contain the same elements. For example, two trajectories such that the state at $t = 1$ of the first is taken as the initial condition of the second. Similarly, in a system for which $\Phi$ is not bijective, two trajectories with different initial conditions may eventually reduce to the same state at the same time. This neither impedes our analysis nor invalidates our notation, with the caveat that neither $i \neq j$ nor $t_a \neq t_b$ guarantees that $x_{i,t_a} \neq x_{j,t_b}$.

[2] This is closely related to the concept of entropy, as it appears in Statistical Mechanics, but further discussion of the topic is beyond the scope of this paper.

## 2.2   Neural Networks

Conventionally, a neural network is given a formulation along the following lines (Schmidhuber, 2015). It is denoted by a function $h : \Theta \times X \to Y$, where $\Theta$ is the space of possible learned parameters, subdivided into the entries of multiplicative weight matrices $W_l$ and additive bias vectors $b_l$. $X$ is the vector space of possible inputs, and $Y$ is the vector space of possible outputs. Each of the $L$ layers in the neural network is given by a matrix multiplication, a bias addition, and the application of a nonlinear activation function $\sigma$, with hidden states $z_{i,l}$ representing the intermediate values taken during the inference operation:

$$z_{i,0} := x_i$$

$$z_{i,l+1} = \sigma(W_l z_{i,l} + b_l) | W_l, b_l \subset \theta$$

$$h(\theta; x_i) = \hat{y}_i := z_{i,L}$$

Without loss of generality, we may transcribe this formulation as a dynamical system by taking its components as analogues. The first is $[L] = \{0, 1, 2 \ldots L\}$, which here will be used to represent the current depth of the hidden state, from 0 for the initial condition up to $L$ for the eventual output. Because it progresses forward during the inference operation, and is associative insofar as increases in depth are additive, $[L]$ functions as an analogue for $T$. The second is $Z$, which is the vector space of all possible hidden states, and thus replaces $X$. The final component is $g : [L] \times Z \to Z$, which here we will write as

$$z_{i,l+1} = g(1, z_{i,l}) = \sigma(W_l z_{i,l} + b_l)$$

A further discussion of the function $g$ is given in Appendix A. The generalization to $g(\Delta l, z_{i,l})$ then follows from the same rule of composition applied to the dynamical systems, at least for integer values of $\Delta l$, under the condition that it never leaves $[L]$. This allows us to replace $\Phi$ with $g$. We can also then re-write the notation along the lines of that for the dynamical systems

$$g(l, z_{i,0}) = g^l(x_i)$$

Noting of course that we have defined $z_{i,0}$ as $x_i$. Thus, the neural network inference operation can be rewritten as the triplet $([L], Z, g)$, and mapped to the dynamical system formulation of $(T, X, \Phi)$. We can now start to discuss the trajectories of the hidden states of the neural network, and what happens when their inputs are changed slightly. For the first hidden state, defined as the input, we can immediately say that

$$|g^0(x_i) - g^0(x_i + \delta x)| = |\delta x|$$

and then by once again mapping to the dynamical systems perspective, we model the difference between the two trajectories at depth $l$ with

$$|g^l(x_i) - g^l(x_i + \delta x)| \approx |\delta x| e^{\lambda l}$$

While, as per the dynamical system, using an exponential model is typically the most illustrative despite the growth not necessarily being exponential, a basic theoretical justification for an exponential model is provided in Appendix B. Continuing, when the value of $\lambda$ is greater than 0, we may call the neural network sensitive to its input, in precisely the same manner as a dynamical system is sensitive to its initial conditions. We may also say that, when the value of $e^{\lambda L}$ is very large, it being the ratio of the magnitude of the change of the output to the magnitude of the change in the input, $\delta x$ becomes an adversarial perturbation. If this analogy holds, we should expect that when we adversarially attack a neural network, the difference between the two corresponding hidden states should grow as they progress through the model. This is our first experimental result.

As an aside, there is a tangential connection to be made between the Chaos Theoretic formulation of neural networks and Algorithmic Stability, like that discussed in Kearns & Ron (1997); Bousquet & Elisseeff (2000; 2002) and Hardt et al. (2016). However, while Algorithmic Stability also treats a notion of the effects of small changes in Machine Learning models, this is from the perspective of changes being made to the learning problem itself, such as to the training dataset, and the resulting effects on the learned model, rather than the effects of small changes being made to individual inference inputs and their respective outputs once the model has already been produced.

## 3 Experimental Design

For our experiments, we used two different model architectures: ResNets (He et al., 2015), as per the default Torchvision implementation (Marcel & Rodriguez, 2010), and a basic custom CNN architecture in order to have finer-grained control over the depth and number of channels in the model. The ResNets were modified, and the custom models built as to allow for recording all of the hidden states during the inference operation. These models, unless specified that they were left untrained, were trained on the Food-101 dataset from Bossard et al. (2014) for 50 epochs with a batch size of 64 and a learning rate of 0.0001 with the Adam optimizer against cross entropy loss. The ResNet models used were ResNet18, ResNet34, ResNet50, ResNet101, and ResNet152.

In the Torchvision ResNet class, models consist of operations named *conv1, bn1, relu, maxpool, layer1, layer2, layer3, layer4, avgpool,* and *fc,* with the first four representing a downsampling intake, then four more blocks of ResNet layers, and then a final operation that converts the rank 3 encoding tensor into a rank 1 class weight tensor. Hidden states are recorded at the input, after *conv1, layer1, layer2, layer3, layer4,* and at the output.

The custom models, specified with $C$ and $D$, consist of $D$ tuples of convolutional layers, batch normalization operations, and ReLU nonlinearities, with the first tuple having a downsampling convolution and a maxpool operation after the ReLU. Each of these convolutions, besides the first which takes in three channels, has $C$ channels. Finally, there is a $1 \times 1$ convolution, a channel-wise averaging, and then a single fully connected layer with 101 outputs, one for each class in the Food-101 dataset. Hidden states are recorded after every tuple, and also include the input and the output of the model. The first tuple approximates the downsampling intake of the ResNet models.

In order to better handle the high dimensionality and changes in scale of the inputs, outputs, and hidden states, rather than using the Euclidean $L2$ norm as the distance metric, we used a modified Euclidean distance

$$|\vec{v}| := \sqrt{\frac{1}{\mathbf{dim}(\vec{v})} \sum_i v_i^2}$$

which will be applied for every instance of length and distance of and between hidden states, including attack radii. Adversarial perturbations $\delta x_{adv}$ against a neural network $h(\theta; \cdot)$ of a given radius $r$ for a given input $x_i$ were generated by using five steps of gradient descent with a learning rate of 0.01, maximizing

$$|h(\theta; x_i) - h(\theta; x_i + \delta x_{adv})|$$

and projecting back to the hypersphere of radius $r$ after every update. These attacks closely resemble those in Zhang et al. (2021); Wu et al. (2021b; 2022); Xie et al. (2021) and Shao et al. (2021), and their use of attacks with $l_p$-norm decay metrics or boundaries. For comparison, random perturbations were also generated, by projecting randomly sampled Gaussian noise to the same hypersphere. In order to perform these experiments under optimal conditions, the inputs that were adversarially perturbed were selected only from the subset of the Food-101 testing set for which every single trained model was correct in its estimate of the Top-1 output class. A Jupyter Notebook implementing these training regimes and attacks will be made available alongside this manuscript, pending review.

## 4 Hidden State Drift

An example of the approximately exponential growth in the distance between the hidden state trajectories associated with normal and adversarially perturbed inputs hypothesized in section 2.2 for 32 inputs is shown in Figure 2. Between the initial perturbations, generated with a radius of 0.0001, and the outputs, the differences grew by a factor of $\sim 747 \times$. Given that ResNet18 has 18 layers, using $747 \approx e^{18\lambda}$, we can calculate $\lambda \approx 0.368$, an average measure of this drift per layer. However, the Lyapunov exponent for each layer is of less interest to an adversarial attacker or defender, with the actual value of interest being given by this new metric, $\psi$, the adversarial susceptibility for a particular input and attack, given by

$$\psi(h, \theta, x_i, \delta x_{adv}) := e^{\lambda L} = \frac{|h(\theta; x_i) - h(\theta; x_i + \delta x_{adv})|}{|\delta x_{adv}|} \tag{1}$$

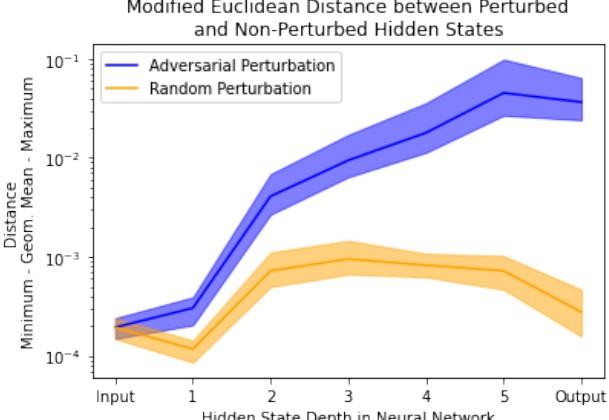

Figure 2: Example of hidden state drift while performing inference with the ResNet18 model. Note the logarithmic scaling on the $y$-axis.

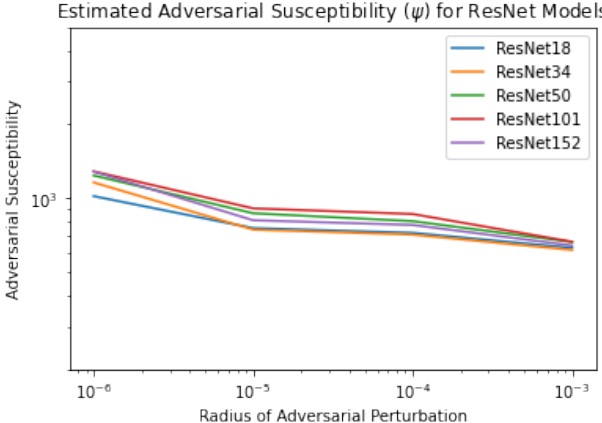

Figure 3: Despite a change in the radius of the adversarial perturbation by three orders of magnitude, the value of $\psi$ associated with those attacks remains relatively stable.

This is the "susceptibility ratio," the ratio of the change inflicted by a given adversarial perturbation to its original magnitude. If this is a meaningful metric by which to judge a neural network architecture, it should remain relatively stable despite changes in the radius of the adversarial attack. This is our second experimental result, demonstrated in Figure 3. By sampling $\psi$ over a number of inputs $x_i$ from a dataset $\mathcal{D}$ and a variety of attack radii $r_{min} \leq |\delta x_{adv}| \leq r_{max}$ and taking the geometric mean[3], we can come to a single value, written as

$$\Psi(h, \theta) = e^{\mathbb{E}_{x_i \sim \mathcal{D}, |\delta x_{adv}| \sim [r_{min}, r_{max}]}[\ln(\psi(h, \theta, x_i, \delta x_{adv}))]} \tag{2}$$

and approximated with $\hat{\Psi}(h, \theta)$, giving the susceptibility ratio for the model as a whole. These values have been calculated for the trained ResNet models, and are given in Table 1. These experimental results are more in line with those of Cazenavette et al. (2020) and Huang et al. (2022b) than with the predictions that we will make in the next section, at which point we will begin using our custom model architectures to tease out the relationships between a neural network's architecture and its susceptibility ratio. A discussion of this metric's relationship to the Lipschitz constant is provided in Appendix C.

---

[3]In order to increase the numerical stability of the geometric mean calculation, we use $\sqrt[n]{\prod_{i=0}^{n} a_i} = e^{\frac{\sum_{i=0}^{n} \ln a_i}{n}}$

| | ResNet18 | ResNet34 | ResNet50 | ResNet101 | ResNet152 |
|---|---|---|---|---|---|
| $\hat{\Psi}(h,\theta)$ | 781.2 | 790.7 | 854.4 | 893.2 | 846.5 |

Table 1: Overall susceptibility ratio of trained ResNet models.

$$\hat{\Psi}(h,\theta)$$

| | | Channels ($C$) | | | |
|---|---|---|---|---|---|
| | | 32 | 64 | 128 | 256 |
| Layers ($D$) | 2 | 0.749 | 0.523 | 0.651 | 0.560 |
| | 4 | 1.021 | 0.695 | 0.775 | 0.610 |
| | 8 | 2.788 | 2.134 | 1.505 | 1.276 |
| | 16 | 15.491 | 12.935 | 8.423 | 7.123 |
| | 32 | 109.340 | 135.472 | 98.834 | 92.404 |
| | 64 | 96.037 | 63.785 | 60.443 | 48.721 |

Table 2: Susceptibility ratio of randomly initialized convolutional models with custom architectures on inputs consisting of random noise.

## 5 Architectural Effects on Adversarial Susceptibility

Returning to the definition of $\psi$ given in equation 1, we might model it as being proportional to the exponent of $L$, the depth of the neural network. Yet, despite ResNet152 having more than eight times as many layers as ResNet18, its susceptibility is only marginally higher. This effect was explored to a greater experimental degree in Cazenavette et al. (2020) and Huang et al. (2022b), demonstrating a remarkable tendency towards robustness in residual model architectures. Interestingly, Huang et al. (2022b) found that deeper models were more robust than wider models, which runs counter to both the experimental and theoretical evidence provided here. Proceeding, this makes the use of an exponential model, at least to explain these experimental results, limited. In order to explore this reasoning further in a more numerically ideal setting, we present our third experimental result, in Table 2, and replicated in Figure 4. Here, using randomly initialized, untrained models with custom architectures as described in the experimental methods section (3), having them perform inference on random inputs, and then adversarially attacking the same models on the same inputs, we can tease apart the relationship between model architecture and the resulting susceptibility ratio, for the case where both parameters and input dimensions are given as Gaussian noise.

We immediately find an approximately exponential relationship between the susceptibility ratio and the depth of the model that was expected based on equation 1, however the slight dip upon moving from 32 to 64 layers is unexpected, and while exploring its potential causes and implications is outside of the scope of this paper, it may warrant further experimentation and analysis.

Also of interest is the effect, or lack thereof, of increasing the number of channels in the neural network. While a quadratic increase in the number of parameters in the model might be expected to increase its susceptibility ratio, especially per the theoretical analysis in Huang et al. (2022a), no experiment that we performed yielded such a result. Some theoretical analysis and discussion is provided in Appendix B.

We repeated the susceptibility ratio measurement on the same model architectures, this time with trained parameters, again sampling the inputs from the Food-101 testing subset for which all models produced correct Top-1 class estimates. These results are in Table 3, and replicated in Figure 5.

The largest resulting difference is the increase of susceptibility for every model by multiple orders of magnitude. Training the models and switching to an information-rich input domain has resulted in trained models being far more sensitive to attack. Yet, following earlier experiments, we can again see that the number of channels has minimal and unclear effects on the susceptibility ratio of the model, while the number of layers increases it significantly. However, for these experimental results, the relationship between the number of layers and the susceptibility has changed, more closely resembling logarithmic than exponential growth, and somewhat replicates the relationship found between depth and susceptibility among the trained ResNet

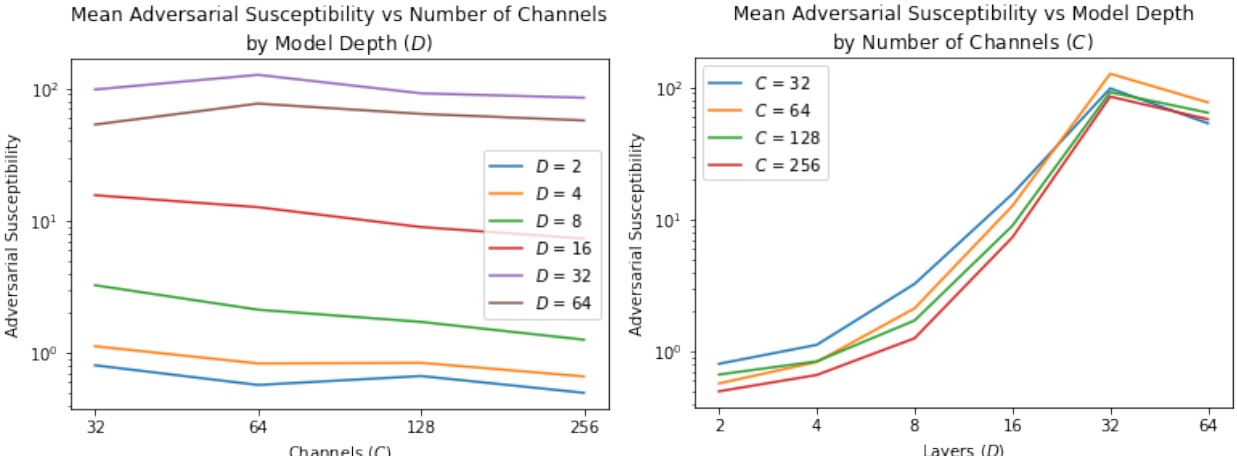

Figure 4: Graphical replication of Table 2; susceptibility ratios of models with random weights.

$$\hat{\Psi}(h, \theta)$$

|  |  | Channels ($C$) | | | |
|---|---|---|---|---|---|
|  |  | 32 | 64 | 128 | 256 |
| | 2 | 578.602 | 610.207 | 586.503 | 576.759 |
| | 4 | 1470.399 | 1658.209 | 1631.561 | 1695.993 |
| Layers ($D$) | 8 | 2144.418 | 2224.467 | 2536.745 | 2370.648 |
| | 16 | 2361.251 | 2401.381 | 2485.030 | 2846.418 |
| | 32 | 3162.568 | 3018.758 | 2987.640 | 3256.967 |
| | 64 | 2045.765 | 2213.575 | 3103.335 | 2471.823 |

Table 3: Susceptibility ratios of trained custom convolutional models on inputs sampled from Food-101.

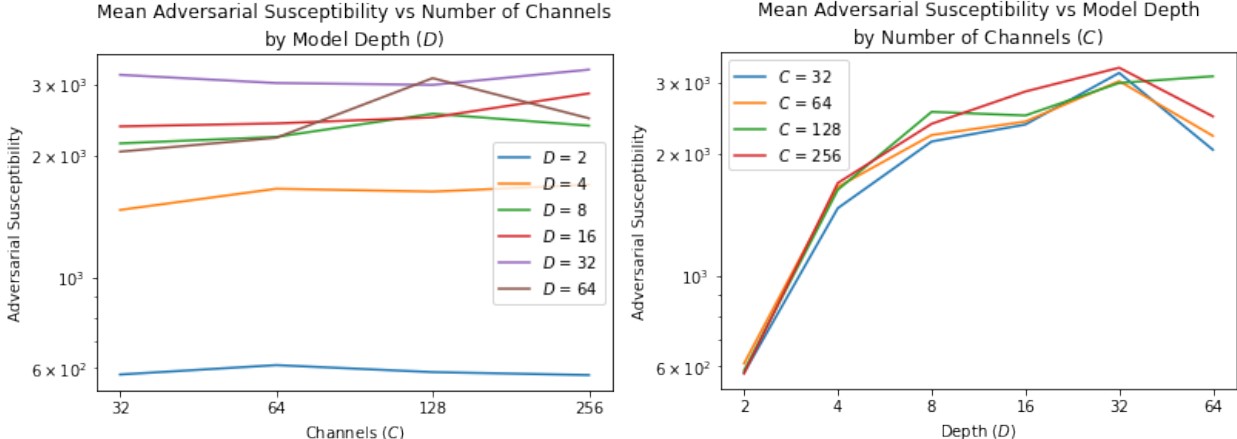

Figure 5: Graphical replication of Table 3; susceptibility ratios of trained models.

models. Interestingly, this is reasonably analogous to the testing accuracy of the models, for which increases in depth yield diminishing returns, and it may be theorized that both of these effects are due to changes in the encoding of information based on model architecture. However, it must be noted that the increase in susceptibility is greater than the increase in accuracy. Making models deeper makes them more vulnerable faster than it makes them more accurate, with additional costs in memory, runtime, and energy consumption.

# 6 Relationships to Other Metrics

## 6.1 Approximation of Certified Robustness Radii

In the work of Weber et al. (2020) and Li et al. (2020), they attempt to calculate what they refer to as "certified robustness radii." For a model with hard decision boundaries, e.g. a top-1 classification model, its certified robustness radius is the largest value $\epsilon_h$ such that, for any input $x_i$ and any adversarial perturbation $|\delta x_{adv}|$, the ultimate classification given by the model $\text{argmax}_c h(\theta; x_i) = \text{argmax}_c h(\theta; x_i + \delta x_{adv})$ for all perturbations with radius smaller than $\epsilon_h$. In their work, however, they state explicitly that these values are highly demanding to calculate for small models, and computationally infeasible for larger models. However, using the susceptibility ratio for a model, one can quickly approximate this certified robustness radius for even very large models. It is simply the distance to the nearest decision boundary, divided by $\hat{\Psi}(h, \theta)$.

We demonstrate with an example: a five-class model outputs the following weights for a given input, $\hat{y} = \{2.1, 0.6, 0.1, -0.5, -1.1\}$. Thus, the nearest decision boundary occurs where the first and second classes become equal, at $\hat{y}' = \{1.35, 1.35, 0.1, -0.5, -1.1\}$. The modified Euclidean distance between these two vectors is 0.4703. Suppose that this model has a susceptibility ratio of $\hat{\Psi} = 25.0$. Its certified robustness radius would then be estimated as $\hat{\epsilon}_h = \frac{0.4703}{25.0} = 0.01897$. One could then take the mean or minimum over these values for every input in a dataset, and a number could be produced for the model as a whole. It must be noted that this will produce a substantial overestimate of the actual certified robustness radius, as the susceptibility ratio is a geometric mean rather than a supremum, and is produced via experimental approximation rather than a numerical solution. However, this "approximated robustness radius" is also useful in practice, as it provides a much larger radius wherein the associated model is highly probably immune from attack, rather than an extremely small radius wherein the associated model is provably immune from attack.

Finally, an over-all criticism has to be made regarding the use of these certified robustness radii in general. Consider two models used for a binary classification problem, inferring on the same input, which has been perturbed by adversarial attacks of equal radii. The first model, moving from the vanilla to the adversarial input, changes its output from $\{0.9, 0.1\}$ to $\{0.6, 0.4\}$. The second model, under the same conditions, changes its output from $\{0.55, 0.45\}$ to $\{0.45, 0.55\}$. Using a certified robustness radius, it would be concluded that the first model is the more robust, while a more direct reading of the change in probabilities would declare the second model to be more robust. These certified robustness radii represent a dense and inscrutable encoding of information about both the model and the input distribution, such that it can be difficult to use them as a meaningful metric. Consider as a hypothetical if, in the previous example, the first model produced a highly confident output solely because it was significantly overfit, and the underlying domain of the dataset is non-separable near the input. Although this improves its robustness radius, it makes it more susceptible to attack in the field.

## 6.2 Post-Adversarial Accuracy

One of the existing standard measures of Adversarial Robustness is to measure the accuracy of models on adversarially perturbed inputs. If our analyses and experimental results thus far are correct, we should see an inverse relationship between measured susceptibility ratios and the post-adversarial accuracy for any given attack radius. This is our fourth experimental result, shown in Figure 6. In it, we observe that among ResNets, which possess similar values of $\hat{\Psi}(h, \theta)$, post-attack accuracies are close between models, with an approximate but minor correspondence between higher susceptibilities and lower post-attack accuracy. We also observe, among the custom architectures, represented in Figure 6 by the subset of models with 32 channels and in their entirety in Figure 4, a very close inverse relationship between higher susceptibility

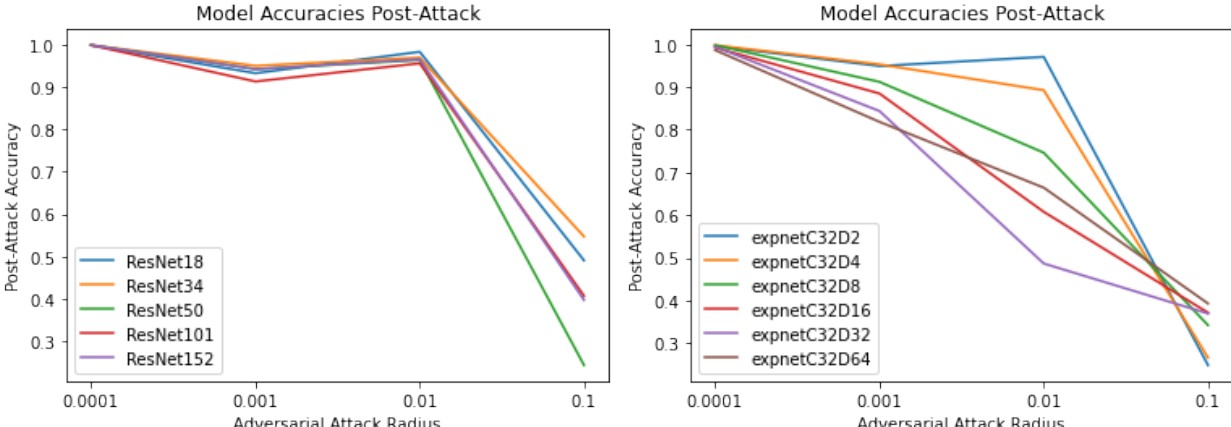

Figure 6: Post-Attack Accuracies

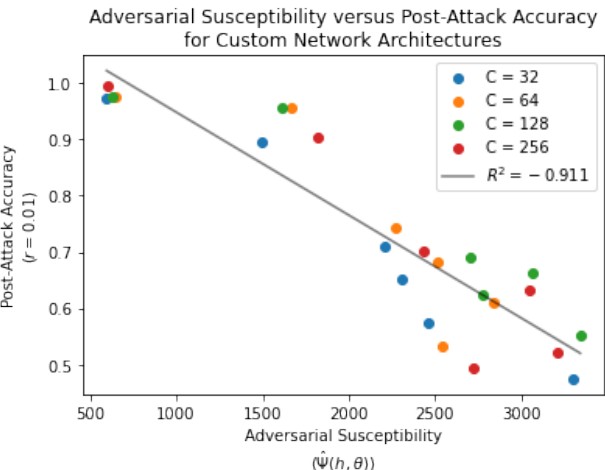

Figure 7: Relationship between Adversarial Susceptibility and Post-Attack Accuracy, with a radius of 0.01. Linear best fit shown, with a correlation coefficient of -0.911.

and lower post-attack accuracy, particularly at the 0.01 attack radius. We also observe that the custom architecture with $D = 2$, which experimentally had $\hat{\Psi} = 578.602$, has a post-attack accuracy curve that closely resembles those of the ResNet models, each of which had a similar susceptibility.

## 7 Conclusions and Future Work

Our experiments have shown, with some variation due to the inscrutable black-box nature of Deep Learning, that there is an extremely strong, analytically valuable, and experimentally valid connection between neural networks and dynamical systems as they exist in Chaos Theory. This connection can be used to make accurate and meaningful predictions about different neural network architectures, as well as to efficiently measure how susceptible they are to adversarial attacks. We have shown a correspondence, both experimentally and analytically, between these new measurements, and those developed in prior works. Thus, a new tool has been added to the toolbox of practitioners looking to make decisions about neural networks.

Future work will include further exploration in this area, and the utilization of more advanced techniques and analysis from Chaos Theory, as well as the development of new, more precise metrics that may tell us more about how models are affected by adversarial attacks. Additionally, the relationship between the

susceptibility ratio and Adversarial Robustness training regimes deserves study, as well as the relationship with different attack methodologies.

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

## A The function $g$

Because the function $g$ only takes $\Delta l$ and $z_{i,l}$ as inputs, and not $l$ itself, a modification must be made in order to define $g$ such that it correctly performs the neural network layer layer operation, while still preserving the same formulation as the evolution function $\Phi(\Delta t, x_{i,t})$. This can be achieved by replacing $Z$ with $Z'$, such that

$$Z' := Z \times [L] = \left\{ \forall (z_{i,l}, l) \big| \exists l \in [L] \wedge \exists z_{i,l} \in Z \right\}$$

Then, $g$ is replaced with $g' : [L] \times Z' \to Z'$, with

$$g'\big(1, (z_{i,l}, l)\big) := \big(\sigma(W_l z_{i,l} + b_l), \ l+1\big)$$

Drawing the index of the parameters $W_l$ and $b_l$ to use from the second element of the $(z_{i,l}, \ l)$ tuple, which it iterates, and with $g'^l$ then following from recursion. This has no bearing in practice, but helps to align theoretical analysis.

# B  Theoretical Basis for Exponential Growth

The use of exponential growth to describe sensitive dependence in Chaos Theory is primarily a model rather than a theoretical result, owing to the typically bounded nature of state spaces. A classic Physical example of a chaotic system is the double pendulum (Levien & Tan, 1993), with a state space defined by the set of possible arm angles $X := [0, 2\pi) \times [0, 2\pi)$, and therefore a maximum $L1$ distance of $2\pi$, bounding exponential growth. For a purely Mathematical example of a chaotic system, consider the state space $[0, 1)$ with the evolution function $\Phi(1, x) := 2x \mod 1$. The trajectory with an initial condition at $x = 0.37$ goes 0.74, 0.48, 0.96, 0.92, 0.84, *et cetera*. Starting with $x = 0.38$, it goes 0.76, 0.52, 0.04, 0.08, 0.16, *et cetera*. The distance between the two trajectories is bounded at 0.5, but starting with a distance of 0.01, it grew to 0.32 in only 5 time steps, for this period having a Lyanpunov exponent of $\ln(2) = 0.693$; positive, and therefore chaotic. With this caveat in mind, a neural network can, to a finite degree and for a temporary period, be expected to produce exponential growth in the hidden state drift of two similar inputs.

## B.1  Random Matrices

Consider a first-order approximation of a neural network which removes the nonlinear activation functions and biases, rendering it a product of matrices. Let us define these to be $d \times d$ real-valued square matrices $W_l \in \mathbb{R}^{d \times d}$, and let us make the simplifying assumption that these are random matrices with i.i.d. univariate Gaussian entries $w_{l_{ij}} \sim \mathcal{N}(\mu = 0, \sigma^2 = 1)$. Next we define a product accumulator matrix $H_L := \prod_{l=0}^{L} W_l$ with elements $\eta_{L_{ij}}$. We will also rely on the following: summing distributions sums their means and variances, and the product of two zero-mean distributions has a variance equal to the product of the variances of its constituents.

For the first two weight matrices $W_1$ and $W_0$ with elements $w_{1_{ij}}$ and $w_{0_{ij}}$, and defining

$$\eta_{1_{ij}} = \sum_{k=1}^{d} w_{1_{ik}} w_{0_{kj}}$$

we get that $\sigma^2_{\eta_{1_{ij}}} = d$ and $\mu^2_{\eta_{1_{ij}}} = 0$. Taking the recursion $H_{L+1} = W_{L+1} H_L$, we get

$$\eta_{L+1_{ij}} = \sum_{k=1}^{d} w_{L+1_{ik}} \eta_{L_{kj}}$$

with $\sigma^2_{\eta_{L+1_{ij}}} = d\sigma^2_{\eta_{L_{ij}}}$ and $\mu^2_{\eta_{L+1_{ij}}} = 0$. Trivially, this resolves to $\sigma^2_{\eta_{L_{ij}}} = d^L$, additionally giving us a standard deviation $\sigma_{\eta_{L_{ij}}} = \sqrt{d^L} = d^{L/2}$. This reduces the accumulator matrix parameter distribution to $\eta_{L_{ij}} \sim \mathcal{N}(\mu = 0, \sigma^2 = d^L)$, and the multiplication of a vector $x_s$ by $H_L$ becomes multiplication by a univariate Gaussian random matrix, and then by $d^{L/2}$, given by $d^{L/2} W_0 x_s$. Substituting out $x_s$ for $x_s + \delta x$, this gives us $d^{L/2} W_0 (x_s + \delta x)$, and finally $H_L x_s - H_L (x_s + \delta x) = -H_L \delta x$, thus

$$\frac{|H_L x_s - H_L (x_s + \delta x)|}{|\delta x|} = d^{L/2} = e^{\ln(d) L / 2}$$

which is an exponential increase in the distance between the trajectories of $x_s$ and $x_s + \delta x$, returning to the earlier dynamical systems formulation.

## B.2  Activation Function

If we may assume that we know the vector $x_s$ beforehand, inserting ReLU activations between each matrix multiplication becomes equivalent to substituting a 0 for each entry in a vector that is being sequentially multiplied by each matrix, itself being equivalent to preserving the vector and instead substituting a 0 for each entry in the associated columns. Because all of the Gaussian distributions discussed have been zero mean, and the probability of a Gaussian being greater than or less than its mean is always 0.5, this gives us,

relying on positive/negative symmetries, that each entry in the resulting vector may equivalently be sampled as

$$x_{s,l_i} = \sum_{k=1}^{d} \sim w_{l_{ik}} x_{s,l_k} \cdot \text{Bern}(p = 0.5)$$

which has the effect of halving the number of dimensions that contributed to $x_{s,l_i}$, in essence lowering $d$ to $d/2$, and further decreasing the growth of the trajectory distance from $e^{\ln(d)L/2}$ to $e^{\ln(d/2)L/2}$.

### B.3  Batch Normalization

Consider a set of vectors that all have some existing magnitude and per-dimension standard deviation from one another. After a normalization step which subtracts out their mean and divides per-dimension by the standard deviation, the resulting vectors will have a mean of 0 and a standard deviation of 1. This resets any growth that they may previously have had away from one another. If a set of vectors including $x_{i,0}$ has a standard deviation of 1, and each element thereof is multiplied by $W_0$ with a ReLU activation applied, such that the set of vectors that includes $x_{i,1}$ has a standard deviation of $\sqrt{d/2}$, the normalization step will result in division by that factor. The effect of this is to curtail the spatially infinite growth of each vector as each neural network layer is applied. Trajectories will still diverge from one another, and the dynamics of the underlying system is such that small changes will still compound, e.g. one entry with a value of 0.01 will have propagating and compounding effects compared to an entry with a value of -0.01, which will simply be erased by ReLU. But it places restrictions on the otherwise infinite possible growth, much like the domain boundaries of the double pendulum or the $\Phi(1, x) := 2x \mod 1$ system discussed earlier.

## C  Adversarial Susceptibility and the Lipschitz Constant

The Lipschitz constant M (European Mathematical Society, 2023) is defined for a function $f : X \to Y$ as

$$M(f) := \sup_{(x_i, x_j) \in X \times X | x_i \neq x_j} \frac{|f(x_i) - f(x_j)|}{|x_i - x_j|}$$

This possesses a close ontological relationship in formulation to the susceptibility ratio, as defined in equations 1 and 2. They both describe a rate of change of a function's output with respect to its input. However, while this relationship is obvious, there are several points of differentiation. The susceptibility ratio is a numerically estimated geometric mean, whereas the Lipschitz constant is a global maximum, which cannot be produced analytically for Deep neural networks. Additionally, whereas the Lipschitz constant is a tool of Analysis, the susceptibility ratio is a construct of combining the Chaos Theoretic Lyanpunov exponent with a constant time horizon.

