# OpenReview forum: "Chaos Theory and Adversarial Robustness"
_TMLR — Rejected by TMLR_

### Review · Reviewer_rpfA · 2023-05-23

**Summary Of Contributions:**

This paper studies the adversarial attacks on deep neural networks with the Chaos Theory. The "time" factor in Chaos Theory can be seen as the depth of the neural network, and the "state space" is the hidden representation. Based on the analogy, a small adversarial perturbation could result in a larger shift in the output following the exponential growth of the depth. The experimental results confirm this analogy under the condition that input samples have been correctly classified by a set of models. Based on this finding, this paper proposed a metric called Susceptibility Ratio to estimate how susceptibility a neural network can be attacked.

**Audience:**

Yes

**Broader Impact Concerns:**

No ethical implications.

**Claims And Evidence:**

No

**Requested Changes:**

See the weaknesses part.

**Strengths And Weaknesses:**

Strengths

- The Chaos Theory is well explained, which allows readers could easily follow. The analogy to neural networks is also very clear.


---

Weaknesses

- The paper claims that the susceptibility ratio approximates the certified robustness radii but does not demonstrate any empirical evidence. The proposed metric is an empirical metric. It would be great to provide evaluations that compare this metric with certified robustness radii with smaller models in which certified radii can be computed.
- The formulation of the susceptibility ratio is very similar to the empirical Lipschitz constant used in [1,2]. The difference seems to be the arithmetic mean v.s. geometric mean. It would be great to provide further clarification of the difference and justifications of why the susceptibility ratio is preferable.
- All experiments are conducted with the ResNet family of architectures. It would be better to include other types of models, such as VGG, DenseNet, and ViT. All experiments are evaluated on Food101, and it would be better to conduct them on other datasets to show the proposed method a generic.
- The analogy of the Chaos Theory, deep neural networks and adversarial attacks is a hypothesis. The empirical justifications are limited under the condition that the depth only follows the exponential growth with input samples correctly classified with a set of models. It is unclear in Table 1 why ResNet does not follow this rule.
- Continuing with the previous point, it would be better to make it clear with Figures 4 and 5 that one is with random weights and the other with trained width in the caption.
- The analysis is based on randomly initialised models or standard training. What about adversarial training?
- The insights from the Chaos Theory suggest that depth makes the deep models vulnerable towards adversarial attacks. It is unclear how Chaos Theory could help to design more robust models.


[1] A closer look at accuracy vs. robustness. NeurIPS 2020\
[2] Exploring Architectural Ingredients of Adversarially Robust Deep Neural Networks. NeurIPS 2021

---

> ### Author Response · Authors · 2023-05-23
>
> Thanks for the review. Those are all pretty fair criticisms.
>
> 1) "The paper claims..." I can definitely try to get some basic numbers for that. My hypothesis is that the certified robustness radii will be much, much lower than the empirically estimate versions, because theory places tighter bounds than practice, e.g. No Free Lunch is true theoretically, but Adam basically works most of the time.
>
> 2) "The formulation..." I'll see what I can do, sounds good.
>
> 3) "All experiments..." I limited my experiments to two classes of models: ResNets, and purely feed-forward CNNs with no architectural nuance whatsoever, to try and line up the theory and the practicalities a bit better. I'll see what I can do about getting some results for a variety of model architectures though. I used Food101 specifically because it's the standard dataset that most closely analogizes to real-world problem settings, as many other datasets have incredibly low resolutions and incredibly broad domains. If you have any specific recommendations for other datasets to use, I'll hear them out, but I have my qualms about CIFAR-10 for example, and I don't have the computational resources to approach ImageNet.
>
> 4) "The analogy of the..." I think it might be because the later layers of the ResNets have weights of lesser magnitude, but I can try to provide some experimental justification for that. However, I also want to note that the exponential growth with a Lyapunov coefficient in the exponent is something of a heuristic model, even as it appears in Chaos Theory; basically every state space has bounds which means the growth has to cease at some point, killing the exponential growth. I don't see the bounding of the drift due to attack as a weakness of the theory, only that the increase in drift by several orders of magnitude confirms it.
>
> 5) "Continuing with the..." I can add that, sure.
>
> 6) "The analysis is based..." I can do some basic experiments there, sure.
>
> 7) "The insights from..." Honestly I have no idea, besides a general recommendation to make shallower, wider models. But hey, if I had all the answers, it wouldn't be research.
>
> I'll need to reach out to some colleagues to help with 1, hit my workstation computer with a stick until it quits some weird driver issues, and get some more experiments for you. Not totally sure when I'll have the time to do this between work and classes, but I'll see what I can get back to you.
>
> This was good feedback. I appreciate it. Thank you.

---

### Review · Reviewer_mqoP · 2023-05-24

**Summary Of Contributions:**

This paper investigates the problem of adversarial robustness from the standpoint of dynamical systems.  Using their exponential model of hidden state drift, they propose a metric called susceptibility ratio and perform experiments to analyze how this value is influenced by the number of channels, model depth, and radius of adversarial perturbation.

**Audience:**

Yes

**Claims And Evidence:**

Yes

**Requested Changes:**

- [critical] Please clarify novelty of work by comparing to and discussing related works studying the impact of architecture on adversarial robustness
- [critical] Please discuss how adversarial susceptibility differs from Lipschitz constant

**Strengths And Weaknesses:**

Strengths:
- writing is clear
- good experimental scope
- interesting problem

Weaknesses:
- novelty is unclear - adversarial susceptibility definition seems similar to the definition of Lipschitz constant, some discussion/comparison is necessary.  Also missing discussions/comparisons to related works in advML literature that analyze the impact of architecture on robustness such as:
Gowal, Sven, et al. "Uncovering the limits of adversarial training against norm-bounded adversarial examples." arXiv preprint arXiv:2010.03593 (2020).
Huang, Shihua, et al. "Revisiting Residual Networks for Adversarial Robustness: An Architectural Perspective." arXiv preprint arXiv:2212.11005 (2022).
Huang, Hanxun, et al. "Exploring architectural ingredients of adversarially robust deep neural networks." Advances in Neural Information Processing Systems 34 (2021): 5545-5559.
Wu, Boxi, et al. "Do wider neural networks really help adversarial robustness?." Advances in Neural Information Processing Systems 34 (2021): 7054-7067.

---

> ### Author Response · Authors · 2023-05-26
>
> I hadn't previously considered the connection to the Lipschitz constant, which I'll make clear in the next draft. The difference is not necessarily one of formulation, but of derivation and application. Physics only really has just the one Inverse Square Law, but each individual Physical law that obeys the formulation still gets its own name and applications. I'll also add in a discussion of those four papers.
>
> But I would make two notes. One, only two of those papers provide any theoretical justification, Huang et al 2021 and Wu et al 2021, and their theoretical analyses arise from entirely different fields than mine, which is Chaos Theory. What novelty my paper has comes from being the first to make the connection that Chaos Theoretic sensitive dependence, the "Butterfly Effect," *is* adversarial susceptibility. Maybe that has limited practical or experimental implications at the moment, but it's a novel connection, and provides an avenue for further thought and study.
>
> And two, per https://jmlr.csail.mit.edu/tmlr/reviewer-guide.html
>
> > Crucially, it should not be used as a reason to reject work that isn't considered “significant” or “impactful” because it isn't achieving a new state-of-the-art on some benchmark. Nor should it form the basis for rejecting work on a method considered not “novel enough”, as novelty of the studied method is not a necessary criteria for acceptance. We explicitly avoid these terms (“significant”, “impactful”, “novel”), and focus instead on the notion of “interest”. If the authors make it clear that there is something to be learned by some researchers in their area from their work, then the criteria of interest is considered satisfied. TMLR instead relies on certifications (such as “Featured” and “Outstanding”) to provide annotations on submissions that pertain to (more speculative) assertions on significance or potential for impact.
>
> > Here's an example on how to use the criteria above. A machine learning class report that re-runs the experiments of a published paper has educational value to the students involved. But if it doesn't surface generalizable insights, it is unlikely to be of interest to (even a subset of) the TMLR audience, and so could be rejected based on this criteria. On the other hand, a proper reproducibility report that systematically studies the robustness or generalizability of a published method and lays out actionable lessons for its audience could satisfy this criteria.
>
> I believe that my paper satisfies these conditions. I know it's not the greatest paper to ever be written, but I have been told that the connection is interesting by colleagues and friends, and I produce two generalizable insights, one theoretical - that adversarial susceptibility is due to Chaos Theory - and one practical - that deeper networks tend to be more susceptible.

---

### Review · Reviewer_Vkrq · 2023-05-25

**Summary Of Contributions:**

This paper attempts to connect Chaos system with neural networks. The intuition is explained and numerical experiments are conducted.

**Audience:**

Yes

**Claims And Evidence:**

No

**Requested Changes:**

Please

(1) Provide the proof of "g^l(x_i)-g^l(x_i+dx)\approx |dx|e^{lambda l}".

(2) Improve the writing to make the paper more clear and formal.

(3) Please provide some discussions on algorithmic stability.

**Strengths And Weaknesses:**

(1) My major concern is that the statements in this paper need to be supported with formal mathematical proof. While the authors try to map the neural network system with the Chaos system, it is essential to provide proof in order to state "g^l(x_i)-g^l(x_i+dx)\approx |dx|e^{lambda l}", which is the core theory of this paper.

If the authors do not intend to prove the theory, the current experiments are not sufficient to support their claims, and the presentation of this paper should be significantly changed.

(2) Relationship with algorithmic stability. Algorithmic stability (Kearns and Ron, 1999; Bousquet and Elisseeff, 2001, 2002)) is defined as the difference in the output of the algorithm given different inputs. It is used as a metric to evaluate the algorithm (i.e., we want a stable algorithm), and it is closely related to the generalization performance of the algorithm (Hardt et al., 2016).

While the authors of this paper try to connect the Chaos system with neural networks, they may want to discuss further [1] the difference between the Chaos system and algorithmic stability and [2] whether it is possible to connect algorithmic stability with neural networks.

References:

Kearns, M. and Ron, D. (1999), “Algorithmic stability and sanity-check bounds for leave-one-out cross-validation,” Neural computation, 11, 1427–1453.

Bousquet, O. and Elisseeff, A. (2001), “Algorithmic stability and generalization performance,” in Advances in Neural Information Processing Systems, pp. 196–202.

Bousquet, O. and Elisseeff, A. (2002), “Stability and generalization,” Journal of machine learning research, 2, 499–526.

Hardt, M., Recht, B., and Singer, Y. (2016), “Train faster, generalize better: Stability of stochastic gradient descent,” in International Conference on Machine Learning, pp. 1225–1234.

(3) Writing of this paper:

The writing of this paper can be significantly improved in many aspects.

The primary issue is that there is some missing definition/logic that prevents people from understanding the math correctly:

[1] In Section 2.1, what does the subscript i in x_{i,t} mean?

[2] The notation z is sometimes written as z_l and z_{i,l} in other cases.

[3] In Section 2.2, please use the mathematical expression to explain "We will handwave the method by which g... the ordinary operations of the feed-forward layer".

In addition, the writing can be more formal. The current writing style is not formal enough for an academic paper. While the writing is easy to understand, I would suggest the authors polish it again. Below are some examples:

[1] The words "this" and "that" are used too many times in the paper. These expressions make the writing informal when appearing frequently. Please try to remove them.

[2] The introduction at the beginning of Section 2 "We’ll start by ... But we should begin by explaining..." is informal. It can be changed to "This paper aims to ... To begin with, we first introduce the definition of ...".

[3] In Section 2.1, paragraph 1, the expression "or something like it" should be removed. We need a precise definition of ingredient one and avoid vague expressions.

[4] In Section 2.1, paragraph 2, the expression "this has to have some properties like" and "like so" are not formal. The expression "like so" also appear in the later paragraphs as well.

[5] In Section 2.1, paragraph 3, what does it refer by "From this"?

[6] Page 3, paragraph 1, the expression "The final piece of the puzzle" is informal.

[7] Page 3, paragraph 1, the sentence "We then need some notion of the distance between two elements of the state space... and proceed from here" can be changed into "To further quantify the distance between ... we assume a vector space equipped with a metric |.| ..."

[8] Page 3, paragraph 1, the expression "know off the bat" is informal.

[9] Page 3, paragraph 2, the sentence "no matter what, you reach the resting state, and that’s the end of the story" is informal.

[10] Page 3, paragraph 2, the expression "until they might as well have started" is informal.

[11] Section 2.2, paragraph 2, "which is written here as" can be changed to "which is written as".

[12] Section 2.2, paragraph 3, "perhaps using something along" is informal.

Other writing issues:

[1] In the abstract, please simplify the sentence "We also demonstrate ... post-attack accuracy". Its current presentation is too long, and it is hard to capture the logic among all the parts in this sentence.

[2] In Section 2.1, paragraph 1, please clarify the logic between the definition T and the sentence "A change in time can be added to an initial time to get an end time, in an associative fashion." When reading this paragraph, I can understand T, but I'm not sure the purpose of the sentence as mentioned.

[3] It seems like there is one extra line between the first paragraph of Section 2.1 and the formula after it. Similar issues appear in many of the formulas in this paper. Please check through the paper and fix them.

[4] Incorrect use of "e.g.". Please check whether "i.e." or "e.g." is needed.

---

> ### Author Response · Authors · 2023-05-28
>
> Thanks for the insight. I have added an appendix that tries to provide some theoretical basis for the exponential growth of hidden state drift - not a proof, exactly, as even Chaos Theory acknowledges that the growth is typically bounded and that the exponential nature of the growth is typically more of an experimental model or heuristic than an analytic result - on the basis of the distributions of the entries of products of increasingly large numbers of random matrices.
>
> I will also be implementing most of your suggestions regarding prose, and attempting to make the entire manuscript more formal generally.
>
> I have also added a paragraph that discusses the relationship to algorithmic stability, with the primary note being that algorithmic stability is more about the stability of the learning algorithm in the face of changes to the learning problem, not the stability of particular inferences made by an already trained model.

---

### Decision · Action_Editors · 2023-06-25

**Recommendation:** Reject

**Comment:**

This paper connects adversarial robustness to chaos theory. One contribute is to propose a new metric to capture how greatly a model’s output will be changed by perturbations to a given input. The presentation of the paper is not very professional, as pointed out by one reviewer that I agree with. The reviewers also raise some other concerns, for example, the connection with algorithmic stability, the relation between adversarial susceptibility and Lipschitz constant, the need for more empirical evidence, and unsufficient to demonstrate the contribution of the paper. Unfortunately, the author did not provide a good rebuttal to address the issues. Many of the rebuttal are just general acknowledgements without evidence.

To sum up, although the connection between adversarial robustness and chaos theory looks interesting. However, I believe the connection is not sufficiently well established, and the implication is not very clear. Thus, I recommend rejection.

**Audience:**

Researchers in adversarial robustness might find the paper interesting.

**Claims And Evidence:**

The paper tries to connect adversarial robustness to chaos theory. Although the connect looks interesting, the claims are not well supported by clear evidence. See below for more comments.